

# Sensitivity analysis of the infection transmissibility in the UK during the COVID-19 pandemic

Pardis Biglarbeigi, Kok Yew Ng, Dewar Finlay, Raymond Bond, Min Jing and James McLaughlin

Faculty of Computing, Engineering and Built-Environment, University of Ulster, Newtownabbey, County Antrim, UK

## ABSTRACT

The coronavirus (COVID-19) outbreak started in December 2019 and rapidly spread around the world affecting millions of people. With the growth of infection rate, many countries adopted different policies to control the spread of the disease. The UK implemented strict rules instructing individuals to stay at home except in some special circumstances starting from 23 March 2020. Accordingly, this study focuses on sensitivity analysis of transmissibility of the infection as the effects of removing restrictions, for example by returning different occupational groups to their normal working environment and its effect on the reproduction number in the UK. For this reason, available social contact matrices are adopted for the population of UK to account for the average number of contacts. Different scenarios are then considered to analyse the variability of total contacts on the reproduction number in the UK as a whole and each of its four nations. Our data-driven retrospective analysis shows that if more than 38.5% of UK working-age population return to their normal working environment, the reproduction number in the UK is expected to be higher than 1. However, analysis of each nation, separately, shows that local reproduction number in each nation may be different and requires more adequate analysis. Accordingly, we believe that using statistical methods and historical data can provide good estimation of local transmissibility and reproduction number in any region. As a consequence of this analysis, efforts to reduce the restrictions should be implemented locally via different control policies. It is important that these policies consider the social contacts, population density, and the occupational groups that are specific to each region.

Corresponding author
Pardis Biglarbeigi,
p.biglarbeigi@ulster.ac.uk

## INTRODUCTION

The novel coronavirus disease (COVID-19) outbreak began in December 2019 and it has spread quickly with 27.7M confirmed cases and more than 889K deaths worldwide as of 10 September 2020 (*WHO, 2020*). COVID-19 is a severe acute respiratory syndrome that can be transmitted by close human-to-human contact via droplets. Consequently, many countries adopted physical distancing policies in attempt to control the spread of

this infectious disease by reducing social contacts. These interventions in social contact behaviour changed the transmission rates and the reproduction number, $R$, which describes the spread of the disease (*Jarvis et al., 2020*; *Davies et al., 2020*).

The basic reproduction number, $R_0$, is the measure of transmissibility and contagiousness of the infectious disease. $R_0$ is an epidemiology indicator that is considered to be a fundamental indicator in the analysis of infectious disease dynamics (*Pellis, Ball & Trapman, 2012*; *Delamater et al., 2019*). If the $R_0$ value is higher than 1, it is expected that the number of active infections will continue to grow and if it is less than 1, it is expected that the number of active infections will decline (*Diekmann, Heesterbeek & Metz, 1990*). Furthermore, $R_0$ can be used to determine the magnitude of the outbreak which can in turn inform the extent to which population vaccination will be effective (*Anderson & May, 1985*; *Delamater et al., 2019*).

In the UK, the first confirmed case of COVID-19 was reported on 31 January 2020 and since the beginning of the outbreak, there have been more than 358K lab-confirmed cases reported with 41.6K deaths as of 10 September 2020 (*Department of Health, 2020b*). With the rapid growth of the infection and the high value of $R_0$, the UK government implemented a strict physical distancing policy on 23 March 2020, instructing all individuals to avoid contact with others except for essential work, buying essential items, and for one form of exercise a day (*Government of the United Kingdom, 2020a*). With this policy the UK government managed to control the spread of the disease. However, it is vital to investigate what would happen during the ongoing easing of these restrictions. To this extent, epidemiologists call for the reproduction number to be kept below 1 while easing the restrictions. One likely scenario to overcome this obstacle is a 'suppress and lift' strategy which implies that the governments ease the lock-down restrictions where possible and re-implement them when the infection rate climbs up (*Kupferschmidt, 2020a*).

Moreover, in different reports by UK authorities, COVID-19 related death rates by occupation, based on Standard Occupational Classification (SOC) (*Office for National Statistics, 2010*), are published, see Fig. 1. These reports indicate that, in the age range of 20–65, the occupational groups of skilled trades, Elementary, Process, Plant and Machine operatives have the highest COVID-19 related death rates in the UK. Although, it is important to mention that, for this study, we were unable to find adequate data relating COVID-19 death rates among unemployed group in Scotland, England and Wales.

Accordingly, in an expert reaction briefing to these data, published by Science Media Centre (SMC), an expert in Epidemiology and Bio-statistics mentioned that: 'This updated report from ONS strengthens the evidence that, for working age people, COVID-19 is largely an occupational disease' (*Pearce, 2020*).

Accordingly, in the current study, we analysed the effect of easing the restrictions by returning of different occupational groups to their normal work environment on the reproduction number under a sample of exit strategy scenarios. We used the available historical data of confirmed inpatients in the UK to develop a data-driven retrospective analysis on $R_0$. It is worth noting that, according to studies, the mutation of the virus was milder (*Kupferschmidt, 2020b*) as of October 2020 and therefore, considering historical

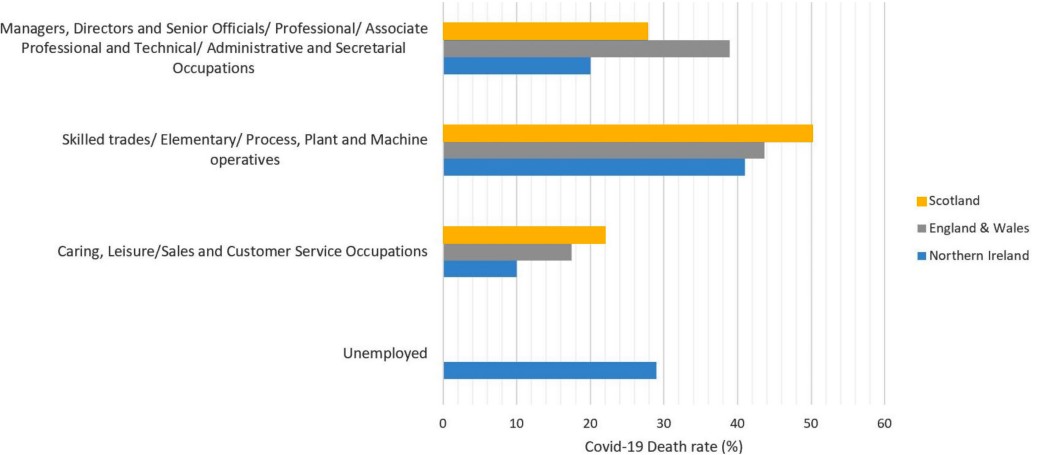

**Figure 1** COVID-19 death rates by occupational groups in the age range of 20–65 (in March, April and May) in Scotland (*National Records of Scotland, 2020*), England & Wales (*Office for National Statistics, 2020a*), and Northern Ireland (*NISRA, 2020*).

data will not result in adequate analysis of death rate or hospital admissions. The proposed analysis in this study considers the social contact matrices and behaviours to calculate the basic reproduction number $R_0$, in the UK. The reproduction number is calculated by applying a sensitivity analysis on the observed daily transmissibility of the infection in the UK. The analysis has further been applied to local population considering the four UK nations separately, i.e. England, Wales, Scotland, and Northern Ireland.

The paper is structured as follows: "Social Contacts" and "Reproduction Number" provide a background to social contact matrices and reproduction number. "Methodology" represents the methodology. "Results and Discussion" provides the results of the study and "Conclusion" represents the conclusion of the current study.

## SOCIAL CONTACTS

In order to understand the transmission dynamics of an infectious disease, social contacts can be studied using contact matrices representing social contact patterns of individuals classified by age groups.

The BBC Pandemic study presents a UK-wide (rural and urban) contact matrix in different contact settings, i.e. home, work, school and other. Overall, this study reported the average contact numbers of about 40.1K participants in 24h from all over the UK (*Klepac, Kissler & Gog, 2018*; *Kucharski et al., 2020*).

However, due to some limitations of the study, further analysis was conducted by *Klepac et al. (2020)* and the average number of contacts, in a 5-year age groups, for weekdays and weekends were presented while considering different contact settings, see Fig. 2. Figure 2C represents that the maximum average number of contacts occur in the educational setting in the age group of 15–19.

Furthermore, it is important to take into account the reciprocal nature of social contacts. This means that the total number of contacts between age group $i$ with age group $j$ must be equal to the total contact numbers of age group $j$ with age group $i$

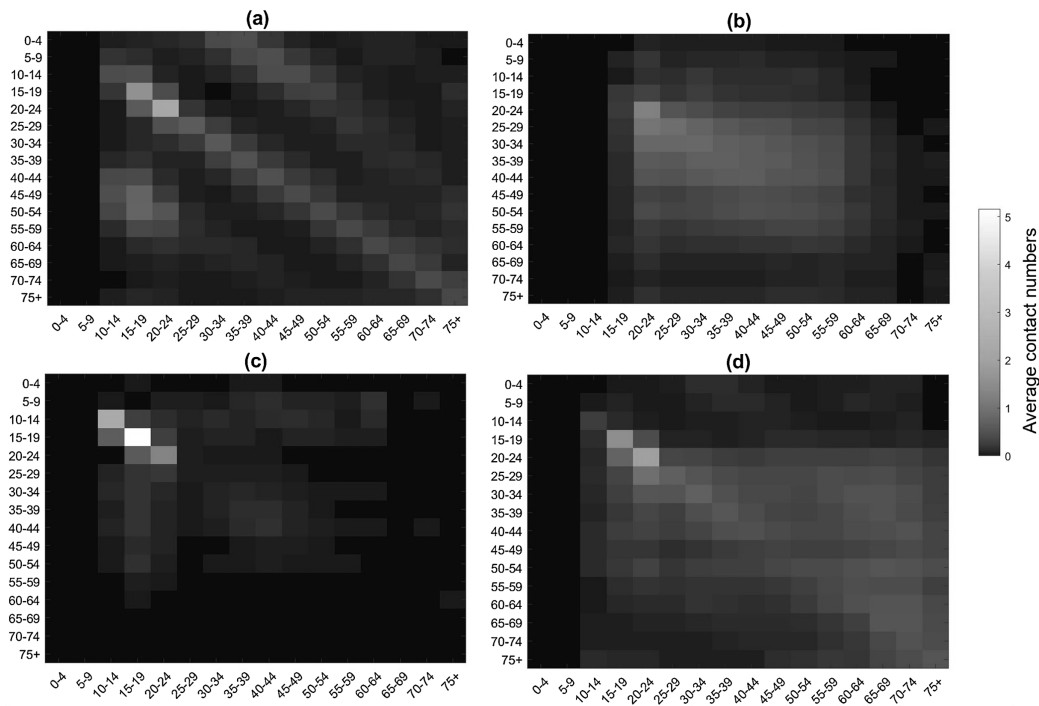

**Figure 2 The BBC Pandemic social contact matrix in different settings of (A) home, (B) work (C) school and (D) other** *Klepac et al. (2020).*

(*Wallinga, Teunis & Kretzschmar, 2006*). Considering $w$ as the total number of individuals in each age group, it is concluded that:

$$m_{ij}w_j = m_{ji}w_i,$$ (1)

Each element of the contact matrix, $m_{ij}$, shows the average number of contacts between the individual of age group $i$ with the individual of the age group $j$ within a unit of time (*Iozzi et al., 2010*). Following this, each element of the reciprocal contact matrix, $C$, can be computed using Eq. (3) (*Klepac et al., 2020*):

$$C = \begin{bmatrix} c_{11} & \dots & c_{1j} \\ c_{21} & \dots & c_{2j} \\ \vdots & & \vdots \\ c_{i1} & \dots & c_{ij} \end{bmatrix},$$ (2)

$$c_{ij} = \frac{1}{2}\left(m_{ij} + m_{ji}\frac{w_i}{w_j}\right).$$ (3)

## REPRODUCTION NUMBER

Social contact patterns affect epidemiological parameters such as the basic reproduction number denoted as $R_0$ (*Wallinga, Teunis & Kretzschmar, 2006*). $R_0$ is an important measure in predicting the severity of a pandemic and is defined as the average/estimated

PeerJ ________________________

number of secondary infections caused by an individual infection in a susceptible population (*Anderson & May, 1992*; *Van den Driessche & Watmough, 2008*; *Tian et al., 2020*). $R_0$ is a dimensionless number which can be calculated as Eq. (4) (*Jones, 2007*):

$$R_0 = \tau \bar{c} d, \tag{4}$$

where $\tau$ is the transmissibility, defined as the probability of infection occurred to a susceptible individual in contact with the primary infector, $\bar{c}$ is the average rate of contacts (contact per time) between susceptible and the infector and $d$ represents the duration of infectiousness which can be defined as one over the recovery rate, $\gamma$. Moreover, contact rate is proportional to the total number of contacts, $T(t)$, in a population size of $N$, over a period of time, $t$ (*Rocklöv & Sjödin, 2020*):

$$T(t) = \bar{c} t N, \tag{5}$$

The value of contact rates, transmissibility and consequently $R_0$ changes over time due to any kind of interventions or control policies. It has been proven that $R_0 < 1$ will lead to a disease-free system (*Ng & Gui, 2020*). Tracking the temporal changes of $R_0$ can help in defining the control measures. Instantaneous reproduction number, $R_t$, is the time-varying reproduction number that shows the average/expected number of secondary infections that might arise from an infection at time $t$ (*Cori et al., 2013*). To this extent, *Thompson et al. (2019)* developed a statistical model, called EpiEstim, to estimate the real-time value for $R_t$ during the outbreak. This widely used model (*Leung et al., 2020*; *Cowling et al., 2020*) considers the serial interval, defined as the interval between the time primary infector and the infected show symptoms (*Wallinga & Teunis, 2004*; *White & Pagano, 2008*), and its corresponding uncertainty. The statistical model of EpiEstim consists of two steps. Initially, the model estimates the serial interval distribution by using the available data on active cases and performs Bayesian parametric estimation on the distribution by data augmentation Markov Chain Monte Carlo (MCMC) to account for the uncertainty of the distribution. The MCMC analysis results in a set of possible values for serial interval distribution which is then used to estimate the $R_t$ value from both the incidence data and the distribution of the serial interval[1] (*Thompson et al., 2019*). Accordingly, EpiEstim estimates the mean and 95% confidence interval of the instantaneous reproduction number. Further, the basic reproduction number can be calculated using (*Gostic et al., 2020*):

$$R_0 = R_t / S(t) \tag{6}$$

where $S(t)$ is the number of susceptible fraction population at time $t$ and is calculated by removing daily deaths and confirmed cases from the total population in each region. The average $R_0$ value obtained from Eq. (6) is reported as the reproduction number.

## METHODOLOGY

In current study, the daily number of active COVID-19 cases, that is confirmed inpatients and outpatients, in all the UK (*Public Health England, 2020*; *Public Health Wales, 2020*; *Public Health Scotland, 2020*; *Department of Health, 2020a*) are used to estimate the

[1] The model is available through an R software package (EpiEstim2.2) and an interactive online interface (EpiEstim App).

instantaneous reproductive number, on a daily basis, using the online EpiEstim App (*Thompson et al., 2019*). EpiEstim starts the simulations with considering $R_t = 2$ and standard deviation of 2. It is then set to simulate 1,000 iterations for each of the distribution parameters via MCMC to account for the uncertainties. For this purpose, multiple published literature are reviewed to define the serial interval (see "Appendix A"), and accordingly, the mean and standard deviation of the serial interval is considered to be 5.9 (95% CI [3.9–9.6]) and 4.8 (95% CI [3.1–10.1]) days, respectively (*Wang et al., 2020a*).

It is acknowledged that the use of different interventions such as universal masking and social distancing measures can effectively reduce the transmissibility of the infection (*Wang et al., 2020b*; *Lan et al., 2020*). However, according to the Science in Emergency Tasking—COVID (SET-C) report (*Mills, Rahal & Akimova, 2020*) in June 2020, although UK has mandated policies on wearing masks in social settings, the practice was not successfully adopted by the public in general, as the uptake of this policy was around 25% compared to 83.4% in Italy, and 63.8% in Spain. Therefore, in this study, it is assumed that the observed daily transmissibility of the infection in each region can be a representative of social behaviour in that region.

Furthermore, the BBC Pandemic contact matrices are used as representative of social contacts in the UK before the start of the COVID-19 outbreak, assuming a homogeneous society, i.e. contacts in all parts of the UK and across all job categories are equally defined by the BBC pandemic social contact matrices. Accordingly, the reciprocal matrices are calculated using the age stratified population size of each nation (*Office for National Statistics, 2020b*). Using the reciprocal contact matrices, total number of contacts are calculated on a daily basis to estimate daily transmissibility, $\tau_t$, according to Eq. (5). In order to consider the effect of the lock-down intervention on transmissibility, the total number of contacts obtained from contact matrices should be modified. For this purpose, with the start of lock-down in the UK on 23 March 2020, contact matrices were modified to include only contacts at home and contacts at work for the percentage of the population who work as health professionals, health and social care associate professionals and those with caring personal service occupations, given the available data from the *Office for National Statistics (2018)*. Moreover, in order to take into account the heterogeneity nature of the contacts in different occupations, the CoMix weekly report data is used (*Jarvis et al., 2020*). CoMix is a social contact survey initiated in 24 March 2020 to report and analyse the changes in contact patterns of individuals in the UK. In CoMix 8[th] report (*Gimma et al., 2020*), it was reported that the average contact number of health professionals, nursing and midwifery professionals, and social work associate professionals is 2.58 times more than the average contact of other professions. Therefore, in this study the average number of contacts obtained by BBC pandemic dataset for the associated percentage of health professionals, in the after lock down settings, is multiplied by 2.58 to account for the increase in their contacts during the pandemic.

Accordingly, the number of contacts in different settings of the BBC contact matrices (i.e. other, schools, and work) after the implementation of the lock-down intervention are considered to be zero. The undertaken assumptions would allow the calculation of daily
observed transmissibility of the infection as the function of confirmed cases and time. Hence, this can be used to represent the social behaviours in each region.

## Exit strategy scenarios

Contact matrices are further modified considering different scenarios, in order to estimate the effect of easing the restrictions by different sectors of labour market returning to normal working environment on reproduction number. The contact matrices are modified to account for the percentage of population with certain type of occupation according to *Office for National Statistics (2018)*. Further they are multiplied by a weight obtained from CoMix survey (*Gimma et al., 2020*), to account for the increase or decrease in the average number of contacts in each scenario. For this purpose, the average daily transmissibility is calculated before and after the lock-down intervention. The average daily transmissibility and total number of contacts from the modified matrices under each scenario are used to estimate the mean value of $R_t$. According to Eq. (4), $R_0$ is further evaluated by considering a value for $d = 1/\gamma$, where $\gamma$ is the recovery rate.

It is worth noticing that many studies have considered very low recovery rate for the UK, for example *Khafaie & Rahim (2020)* has considered a recovery rate of 1.64% by 23 March 2020 in the UK. However, in the current study, due to the lack of available data on the recovery rate in UK, the global recovery rate of 53.38% is considered (*Gondauri, Mikautadze & Batiashvili, 2020*). Therefore, the $R_0$ values can be estimated using the average $R_t$ value defined for different scenarios and the global recovery rate of 53.38%.

Seven scenarios are investigated in this study to illustrate the effect of different sectors of labour market returning to normal working environment on $R_0$ values in the UK. These scenarios are defined as:

SC1 Only skilled trade, elementary occupations and process, plant and machine operatives returning to normal working environment.

SC2 Professional occupations, managers and administrative occupations returning to normal.

SC3 Professional occupations + managers + administrative occupations + skilled trade + elementary occupations + process, plant and machine operatives returning to normal.

SC4 Schools to be opening again + skilled trade, elementary occupations and process, plant and machine operatives and Sales and customer services returning to normal working environment.

SC5 Sales and customer services + leisure and other service occupations returning to normal. However, in this scenario the BBC pandemic contact matrix corresponding to 'other' activities is also considered in estimating the total number of contacts.

SC6 Removing all restrictions with schools remaining closed which account for most of the population returning to normal work environment except individuals with teaching and educational occupations.

SC7 Removing all restrictions.
**Table 1 Percentage of working age population returning to normal working environment under exit scenarios (%) and the associated weight for the average contact numbers ($w$) considered for each scenario.**

| Region | SC1 ($w = 2.35$) | SC2 ($w = 1.30$) | SC3 ($w = 3.33$) | SC4 ($w = 3.43$) | SC5 ($w = 1.08$) | SC6 ($w = 1$) | SC7 ($w = 1$) |
|---|---|---|---|---|---|---|---|
| UK | 24.1 | 38.1 | 62.1 | 38.5 | 14.4 | 96.5 | 100 |
| England | 23.7 | 38.6 | 62.3 | 37.9 | 14.2 | 96.5 | 100 |
| Wales | 27.1 | 34.3 | 61.4 | 42.9 | 15.8 | 96.3 | 100 |
| Scotland | 19.7 | 27.8 | 47.5 | 31.5 | 11.8 | 97.4 | 100 |
| Northern Ireland | 23.5 | 31.0 | 54.5 | 37.3 | 13.8 | 96.7 | 100 |

It is worth noticing that in the UK, each of the nations considered different phases of easing the restrictions at different times, while implementing social distancing measures and other obligations such as wearing masks. However, calculating the number of contacts as well as social behaviour and the effect of social distancing measures in different stages need more detailed investigations. Therefore, in this study, we made the assumption of keeping the contacts to its minimum during the lock-down phase. Accordingly, our calculations show increase in the daily transmissibility due to increase in the number of confirmed COVID-19 inpatients and outpatients. This will eventually affect the $R_0$ value for each of the considered exit scenarios.

Table 1 shows the percentage of the population, in the working age group of 16–74 returning to their normal working environment and the considered weight for the variability of the average number of contacts, based on data presented in *Gimma et al. (2020)*, for each of the considered scenarios. Furthermore, next generation contact matrices can be, retrospectively, calculated using the estimated $R_0$ value for each of the exit scenarios considered. Next generation matrix, *NGM*, represents the spread of infection. Each element of next generation matrix, $NGM_{ij}$, shows the expected number of infections in age group $i$ that has been infected by the individual in age group $j$ (*Diekmann, Heesterbeek & Metz, 1990*; *Diekmann, Heesterbeek & Roberts, 2010*). The dominant eigen value of *NGM* represents the reproduction number, $R_0$ (*Klepac et al., 2020*), therefore:

$$NGM = \frac{R_0}{\rho(C)} C, \tag{7}$$

where $\rho(C)$ is the dominant eigen value of the reciprocal contact matrix, $C$.

Calculation of *NGM*s will allow to determine the transmission of the infection between different age groups, while defining the age-specific risk rate of the infection or 'force of infection'. This can further be used in the time of introduction of the vaccination routine (*Wallinga, Teunis & Kretzschmar, 2006*).

## RESULTS AND DISCUSSION

### $R_t$ and $R_0$ in the UK

The instantaneous reproduction number is calculated using EpiEstim online application for the UK and its four nations from 10 February to 7 September 2020. Figure 3 shows the

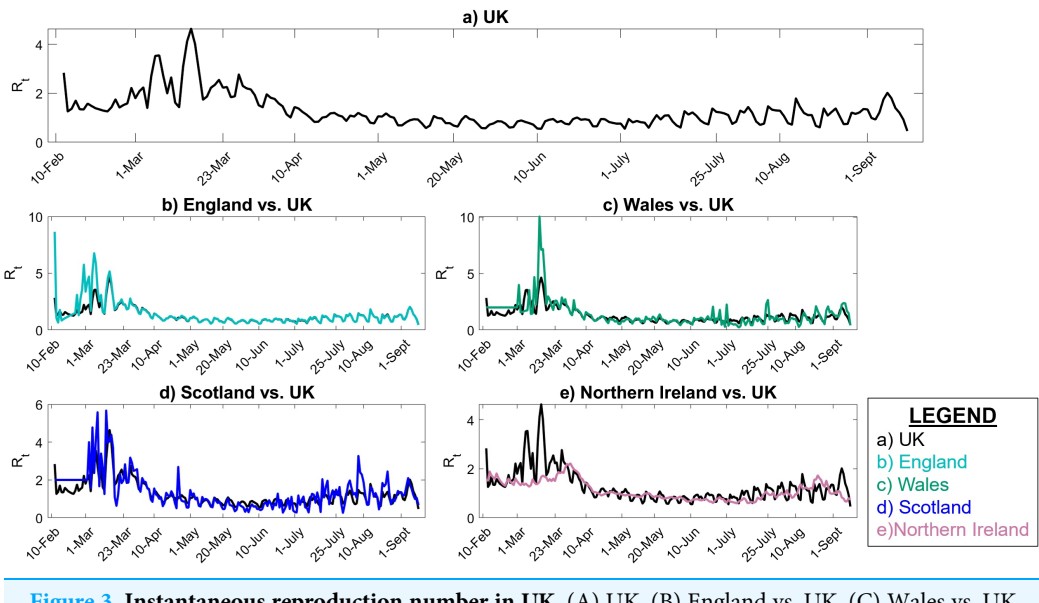

**Figure 3** **Instantaneous reproduction number in UK.** (A) UK, (B) England vs. UK, (C) Wales vs. UK, (D) Scotland vs. UK, (E) Northern Ireland vs. UK.

estimated $R_t$ in the UK and each of its nations compared to UK. As 84.3% of the population of UK resides in England, it is clear that the $R_t$ value in the UK and England are very close. However, the $R_t$ variability in time is slightly different in Scotland and Wales compared to UK, especially during the early stages of the pandemic. In Northern Ireland, $R_t$ has been typically less than the UK as only 2.8% of the UK population resides in Northern Ireland and the population density is lower.

Figure 3 also represents that the $R_t$ value is lower and more steady after the lock-down intervention was implemented by the UK government on 23 March 2020, also stated by *Thompson et al. (2020)*; while, in Wales and Scotland, some spikes of increased $R_t$ values can be seen during the analysis period. Accordingly, the effect of removing the lock-down intervention is being investigated in this paper using contact matrices. The expected $R_0$, considering different exit scenarios, are presented in Fig. 4. Considering SC1, on average about 24% of the UK working age population return to their normal work environment and the average $R_0$ value remains below 1 in all the nations. SC1 shows that Northern Ireland may be less sensitive to the increase in the number of contacts compared to the rest of the other three nations. SC2 considers a higher percentage of population return to their normal working environment, however the associated weight considered for the increase in the average number of contacts is less than SC1, therefore the calculated $R_0$ values are less than SC1 across UK. This suggests that individuals having the occupations considered in SC1 are more likely to spread the infection throughout their daily activities, which also accounts for higher death rates, as shown in Fig. 1.

Due to increased number of social contacts in SC3 to SC7, the average $R_0$ value is higher than 1 in the UK. More detailed values including 95% confidence interval of $R_0$ is presented in "Appendix B".
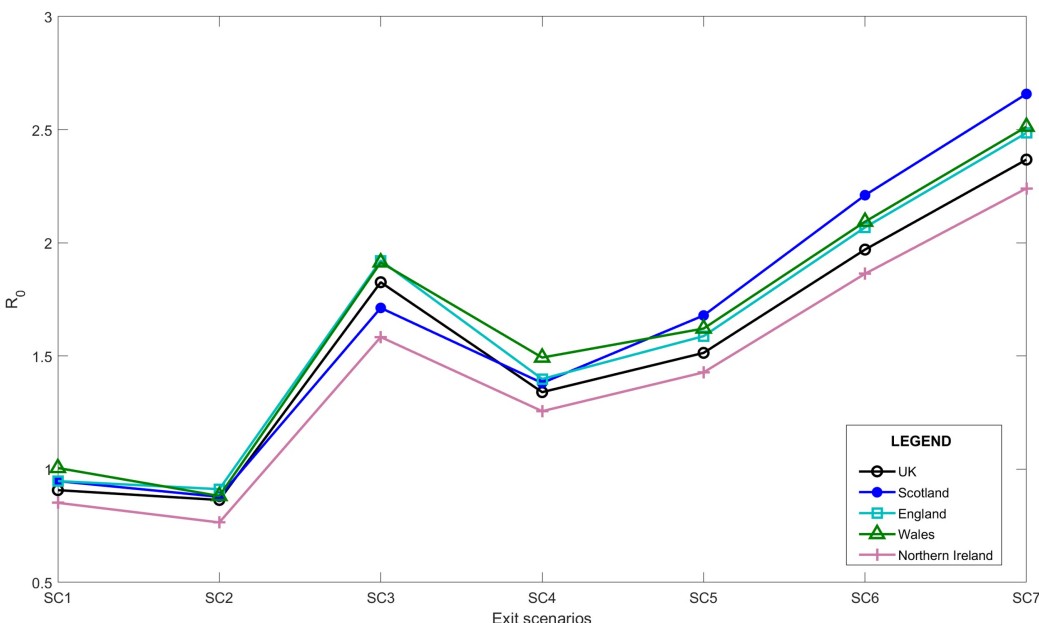

**Figure 4 Expected average $R_0$ after lock-down considering different exit scenarios.**

**Table 2 Expected modelled and estimated $R_0$ in the UK.**

| Region | SC4 (5th–95th%) | Actual estimated | Last updated | Source |
|---|---|---|---|---|
| UK | 1.34 (1.01–1.62) | 1.2–1.5 | 9-Oct-2020 | *Government of the United Kingdom (2020b)* |
| England | 1.40 (0.91–2.51) | 1.2–1.5 | 9-Oct-2020 | *Government of the United Kingdom (2020b)* |
| Wales | 1.50 (0.67–3.34) | 1.3–1.6 | 2-Oct-2020 | *Governement of Wales (2020)* |
| Scotland | 1.38 (0.58–3.08) | 1.3–1.7 | 1-Oct-2020 | *Government of Scotland (2020)* |
| Northern Ireland | 1.25 (0.79–1.79) | 1.5 | 24-Sept-2020 | *UK Department of Health (2020a)* |

As of 10 October 2020, we believe that with the ongoing regulations and policies, the UK is close to SC4 with schools being open. In SC4, we considered a scenario where schools reopen whilst, skilled trade, elementary occupations and processes, plant and machine operatives, as well as sales and customer services return to normal operation modes. In the UK, with the ease of the first national lock-down, schools reopened on 1 June 2020 and have remain open as of 14 Dec 2020. Reopening of the schools account for a substantial increase in the total number of contacts. Table 2 shows the modelled $R_0$ from this study and the actual estimations published for each of the UK nations. According to Table 2, the current sensitivity analysis on the transmissibility can be used to as a base to estimate $R_0$ in the near future.

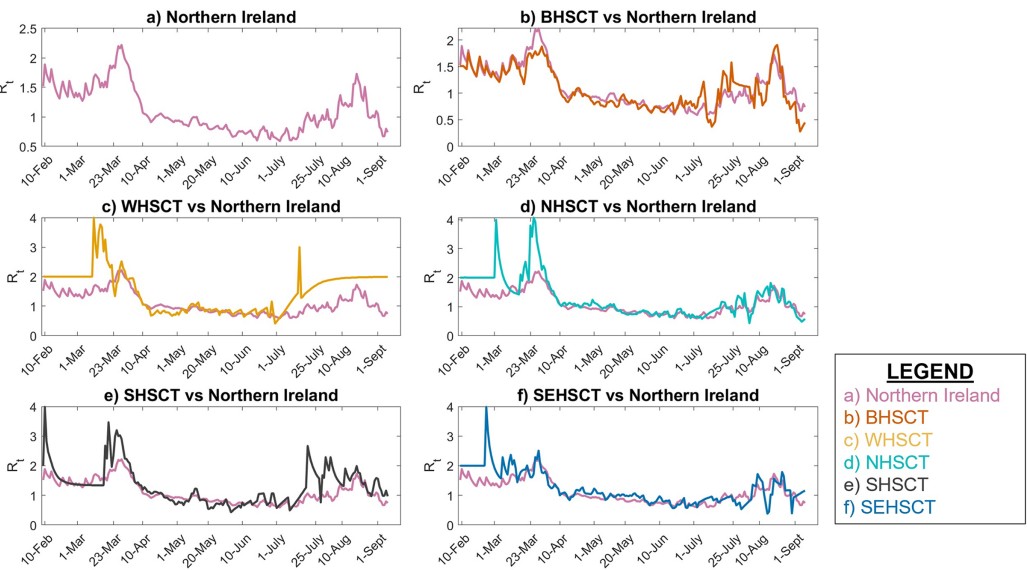

**Figure 5 Local Instantaneous reproduction number in Northern Ireland and its HSC trust areas.**
(A) Northern Ireland, (B) BHSCT vs. Northern Ireland, (C) WHSCT vs. Northern Ireland,
(D) NHSCT vs. Northern Ireland, (E) SHSCT vs. Northern Ireland, (F) SEHSCT vs. Northern Ireland.

## Further analysis for Northern Ireland

Although Northern Ireland shows the lowest values for $R_0$ considering different exit
scenarios, in order to show the importance of the implementation of local $R_0$, we further
analysed its value over Northern Ireland's health trust areas. The healthcare system in
Northern Ireland is divided into 5 Health and Social Care (HSC) trusts, where all the
5 trusts provide an integrated healthcare system throughout the nation that is Belfast HSC
Trust (BHSCT), South Eastern HSC Trust (SEHSCT), Western HSC Trust (WHSCT),
Southern HSC Trust (SHSCT), and Northern HSC Trust (NHSCT). The same retrospective
data-driven analysis is used to estimate the local $R_0$ value for each of the trust areas.

Figure 5 shows that the local $R_t$ in WHSCT, NHSCT, SHSCT and SEHSCT is mostly
higher than Northern Ireland as a whole. This figure also shows that in the final days of the
analysis (1 July–7 September 2020) there is an increase in the $R_t$ value in SHSCT and
WHSCT due to increase in human-to-human contacts as UK entered step 3 of further
easing restrictions on 4 July 2020 (*Institute for Government, 2020*).

Accordingly, our analysis shows that due to transmissibility of the infection and the
working age population size, the $R_0$ values in WHSCT is higher than other HSC trusts in
Northern Ireland with $R_0 = 1.04$ for SC1, although Northern Ireland's $R_0$ value remains
below 1 in this scenario, see Fig. 6. SC2, however, suggests $R_0$ values of lower than 1 in
all the trusts, with WHSCT having $R_0 = 0.94$, which shows a higher sensitivity to changes
in contact numbers compared to other trusts.

SC3 to SC7 represent $R_0$ of higher than 1 in all the regions, suggesting the spread of the
disease again with easing restrictions, if social behaviours remain the same and no further
social distancing policies are considered.

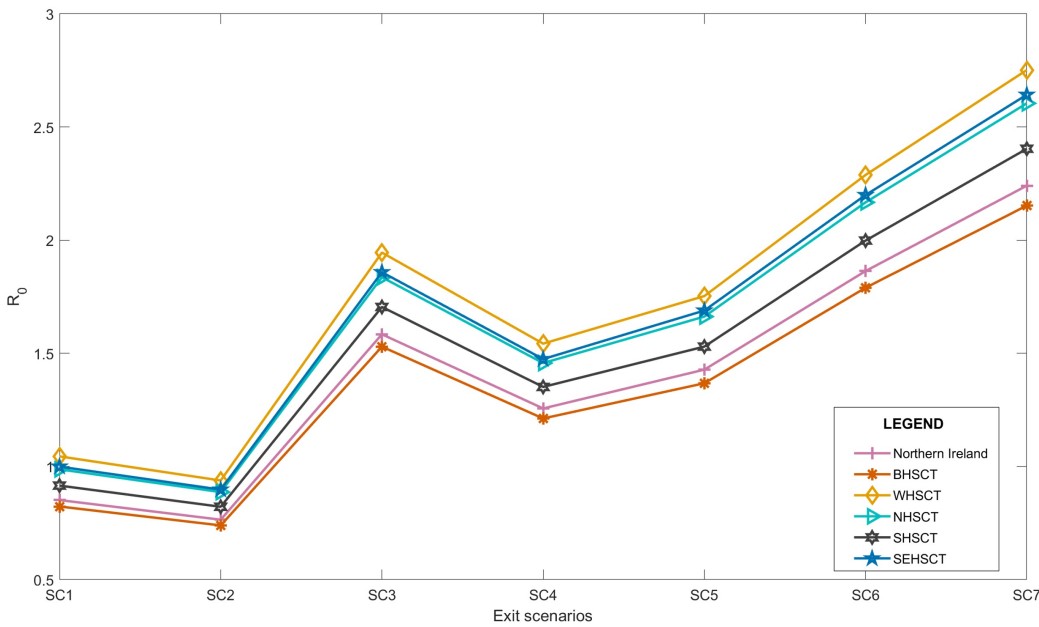

**Figure 6  Local average $R_0$ after lock-down in Northern Ireland, considering different exit scenarios 9/15.**

Moreover, our results show that WHSCT is more sensitive to the number of contacts than other regions in Northern Ireland. According to *Devlin, McKay & Russel (2018)*, WHSCT covers more deprived areas of Northern Ireland, and it is well understood that the low-to-middle income areas require more data collection, investigation, and analysis for adequate policy making (*Thompson et al., 2020*). This is also in agreement with the semi-monthly/monthly reports on the reproduction number published by Department of Health in Northern Ireland (*UK Department of Health, 2020b*).

The sensitivity analysis on transmissibility of infection at local and national levels using historical observed data can produce insights into the prediction of the reproduction number in the near future. This analysis requires limited data, thus enhancing reproducibility. However, assumptions such as considering a weight for average number of contacts in for different occupations can increase the uncertainty of the analysis. Therefore, by considering a more suitable contact matrices, especially at locals level, these uncertainties can be addressed.

Next generation matrix is also calculated using the $R_0$ values for each scenario. Here we only present the NGM results considering the exit scenarios of SC2 and SC7 to illustrate the expected number of new infections by age in the UK, Northern Ireland and WHSCT in Northern Ireland. Figure 7 shows that as the total number of contacts is more in SC7, it is expected that the number of new infections would be higher in all age groups compared to SC2. Also, these figures show that the age-specified expected number of new infections in Northern Ireland is less than UK, however, the results from considering local $R_0$ for WHSCT shows higher number of new cases in this area.

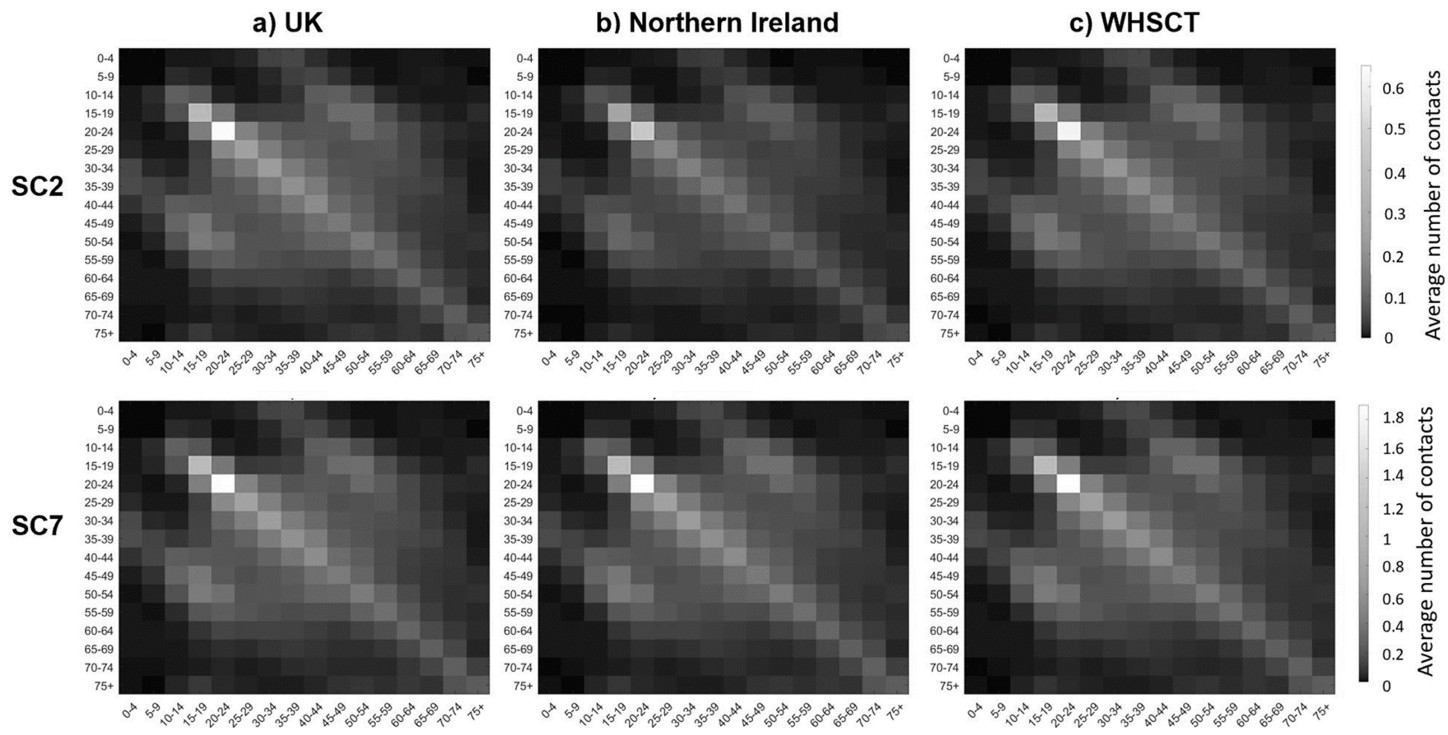

**Figure 7 Next generation contact matrices according to SC2 ans SC7 in (A) UK, (B) Northern Ireland and (C) WHSCT.**

## CONCLUSION

In this study, we performed a sensitivity analysis on transmissibility of COVID-19 infection by considering the effect of returning individuals in different occupations back to the work environment on $R_0$ in the UK and its four nations. The data-driven analysis presented in this study does not model the social distancing measures, it is mainly based on manipulation of number of contacts before COVID-19 started. To this extent, BBC pandemic social contact matrices were adopted to represent average number of contacts in different age groups in the UK. Moreover, $R_0$ values for each of the considered exit scenarios are calculated considering historical transmissibility of the infection in each region, assuming that historical transmissibility of the infection is a representative of the social behaviour in the region.

Our analysis showed that if schools remain open, and furthermore individuals with skilled trade, elementary, process plant, machine operative occupations, sales and costumer services, which account for 38.5% of UK working-age population, return to their normal work environment (SC4) the $R_0$ value is expected to exceed 1. Therefore, it is assumed that each primary infection can cause more than 1 secondary infection. This implies that the infection growth increases and the outbreak will continue to grow.

Although, analysing the social contact data adapted to Northern Ireland shows the lowest value of $R_0$ among the UK nations under all the defined exit scenarios, our study showed that the local $R_0$ values of Northern Ireland's HSC trusts represent higher infectious transmissibility in some areas, which may consequently increase the

**Table 3 Review of published literature on Serial Interval (SI).**

| Reference | Location | Chains | SI Mean (days) | SI SD (days) |
|---|---|---|---|---|
| *Wang et al. (2020a)* | Shenzhen, China | 27 | 5.9 95% CI [3.9–9.6] | 4.8 95% CI [3.1–10.1] |
| *Nishiura, Linton & Akhmetzhanov (2020)* | N/A | 28 | 4.7 95% CI [3.7–6.0] | 2.9 95% CI: [1.9–4.9] |
| *Aghaali et al. (2020)* | Qom, Iran | 51 | 4.55 95% CI [4.27–7.5] | 3.30 95% CI [2–4] |
| *You et al. (2020)* | Hubei, China | 198 | 4.6 | 5.55 |
| *Wu et al. (2020)* | Wuhan, China | 43 | 7.0 (5.8–8.1) | 4.5 (3.5–5.5) |
| *Du et al. (2020)* | China | 468 | 3.96 95% CI [3.53–4.39] | 4.75 days 95% CI [4.46–5.07] |
| *Najafi et al. (2020)* | Iran | 21 | 5.71 | 3.89 |
| *Bi et al. (2020)* | Shenzhen, China | 1,286 | 6.3 95% CI [5.2–7.6] | 4.2 95% CI [3.1–5.3] |
| *Chintalapudi et al. (2020)* | Lombardy, Italy | N/A | 6.6 | 3.1 |

transmissibility across Northern Ireland and may affect the social and health care system as well.

Our study suggests that by relying on the historical data and statistics, the local transmissibility and consequently an estimate of the $R_0$ value can be easily calculated in any region. This knowledge can help in implementing local adaptive control policies in fighting against the growth of COVID-19.

# APPENDIX A

## COVID-19 serial interval

An important key transmission parameter is the serial interval (SI), defined as the time interval between the symptom onset of the infector and infectee. The distribution of serial interval is estimated during the outbreak using the minimum and maximum bands of the symptom onset timing for the primary and secondary cases. This data should be collected during the outbreak by tracking chains of transmission. In any case, there are always uncertainties in the reported timings, specially due to the lack of knowledge in recognising the initial symptom onsets. Therefore, it is essential to consider the uncertainties associated with serial interval (*Thompson et al., 2019*).

In this study, a review of published literature on serial interval distribution (mean and Standard deviation, SD) is presented. Table 3 represents the uncertainty in estimating the serial interval associated with COVID-19. Therefore in this study, the serial interval with the largest confidence interval (CI) is chosen; i.e. SI with mean and SD of 5.9 (95% CI [3.9–9.6]) and 4.8 (95% CI [3.1–10.1]) days, respectively (*Wang et al., 2020a*). In order to estimate the instantaneous reproduction number $R_t$, the EpiEstim online application was set to run 1,000 iterations for each parameter of the distribution via MCMC (*Thompson et al., 2019*) to account for the uncertainties of SI.

# APPENDIX B

## $R_0$ estimation and its confidence interval in the UK

$R_0$ values are calculated considering the 95% confidence interval of instantaneous reproduction numbers, $R_t$, obtained from MCMC simulation of EpiEstim. Tables 4 and 5

**Table 4 Expected average $R_0$ (5th–95th percentile) after lock-down considering different exit scenarios (SC1, SC2, SC3, SC4).**

| Region | SC1 | SC2 | SC3 | SC4 |
|---|---|---|---|---|
| UK | 0.91 (0.68–1.23) | 0.86 (0.68–1.17) | 1.82 (1.38–2.48) | 1.34 (1.01–1.62) |
| England | 0.95 (0.61–1.70) | 0.91 (0.59–1.64) | 1.92 (1.25–3.45) | 1.40 (0.91–2.51) |
| Wales | 0.99 (0.45–2.25) | 0.88 (0.39–1.97) | 1.91 (0.86–4.27) | 1.50 (0.67–3.34) |
| Scotland | 0.95 (0.40–2.11) | 0.88 (0.37–1.96) | 1.71 (0.72–3.81) | 1.38 (0.58–3.08) |
| Northern Ireland | 0.85 (0.53–1.21) | 0.76 (0.48–1.09) | 1.58 (1.00–2.26) | 1.25 (0.79–1.79) |

**Table 5 Expected average $R_0$ (5th–95th percentile) after lock-down considering different exit scenarios (SC5, SC6, SC7).**

| Region | SC5 | SC6 | SC7 |
|---|---|---|---|
| UK | 1.51 (1.14–2.06) | 1.97 (1.49–2.68) | 2.36 (1.79–3.22) |
| England | 1.59 (1.03–2.85) | 2.07 (1.34–3.72) | 2.48 (1.61–4.47) |
| Wales | 1.62 (0.73–3.62) | 2.09 (0.94–4.68) | 2.51 (1.13–5.61) |
| Scotland | 1.68 (0.71–3.74) | 2.21 (0.93–4.93) | 2.66 (1.12–5.92) |
| Northern Ireland | 1.43 (0.90–2.04) | 1.86 (1.18–2.66) | 2.24 (1.42–3.19) |

present the estimated $R_0$ values for the UK, considering the 7 exit scenarios introduced in "Exit strategy scenarios".

### Funding
The authors received no funding for this work.

### Competing Interests
Kok Yew Ng is an Academic Editor for PeerJ.

### Author Contributions
- Pardis Biglarbeigi conceived and designed the experiments, performed the experiments, analysed the data, prepared figures and/or tables, authored or reviewed drafts of the paper, and approved the final draft.
- Kok Yew Ng analysed the data, authored or reviewed drafts of the paper, and approved the final draft.
- Dewar Finlay analysed the data, authored or reviewed drafts of the paper, and approved the final draft.
- Raymond Bond analysed the data, authored or reviewed drafts of the paper, and approved the final draft.
- Min Jing analysed the data, authored or reviewed drafts of the paper, and approved the final draft.
- James McLaughlin analysed the data, authored or reviewed drafts of the paper, and approved the final draft.

## Data Availability

Raw data and code are available in the Supplemental Files.

## Supplemental Information

Supplemental information for this article can be found online at http://dx.doi.org/10.7717/peerj.10992#supplemental-information.

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
