# Peer review of "Sensitivity analysis of the infection transmissibility in the UK during the COVID-19 pandemic"

_PeerJ, doi:10.7717/peerj.10992_

## Round 0.1 · original submission · Major Revisions

Dear authors,

The reviewers have indicated some merit in your work, but there are issues that you must address in a revised version of the text. Please, see the reviewers' comments so as to have more information.

Reviewer 1 ·

Basic reporting

Biglarbeigi et al. used existing data to simulate COVID-19 transmission under different reopening scenario with regard to social contact metrics. The prediction can be helpful for policy making if the algorithm is valid.
In fact, social contact is one of the most important factors associated with R0. However, there are still other factors relating to transmissibility, which further affecting R0; for example, universal masking intervention and social distancing. Existing evidence has shown such intervention can reduce transmissibility even though workers keep working in the pandemic.(1,2) Without taking the effects into account, the predictions may not be accurate.
Second, it is inappropriate to report the results along with the discussion. In “4 RESULTS AND DISCUSSION” section, the authors presented the results, but spared little text on discussion. Therefore there is a lack of interpretations regarding whether the findings are an addition to current science, if it is consistent with previous research, what the underlying mechanisms are, and strengths/limitations of the current study, etc. Without in-depth discussion, it is hard for readers to capture the significance of the study.
Ref.:
1. Wang X, Ferro EG, Zhou G, Hashimoto D, Bhatt DL. Association between universal masking in a health care system and SARS-CoV-2 positivity among health care workers. J Am Med Assoc. 2020;324:703–704.
2. F-Y Lan, C A Christophi, J Buley, E Iliaki, L A Bruno-Murtha, A J Sayah, S N Kales. Effects of universal masking on Massachusetts healthcare workers’ COVID-19 incidence. Occupational Medicine. 2020; kqaa179.

Experimental design

The authors made several assumptions to build the algorithms, but some of them may not reflect the real situation.
1. Line 110-115: the authors counted only “health professionals, health and social care associate professionals and those with caring personal service occupations” as essential workers during the lock-down. However, there are workers in other job categories that need to work despite the mitigation policy, such as transportation, public safety, national security, food and other necessary goods, etc.(3) Without including all essential job categories, the contact matrices estimate for the lock-down period would be incorrect.
2. 3.1 Exit strategy scenarios: To my understanding, the BBC Pandemic contact matrices did not calculate contact for each job category, but for working population as a whole. However, contact rate actually varies across occupations. For example, grocery store workers should have much higher contact rates than administrative workers. Therefore, using the BBC Pandemic contact matrices to estimate each scenario may not be appropriate, as the authors listed particular job categories for individual exit strategy.
3. The paragraph following Line 144, “Accordingly, our calculations show increase in the daily transmissibility due to increase in the number of confirmed COVID-19 inpatients.” I was wondering why the increase in transmissibility was due to “inpatients” but not outpatients or total case numbers. In my opinion, inpatients should be less transmissible than outpatients.
Ref.:
3. The Lancet. The plight of essential workers during the COVID-19 pandemic. Lancet. 2020;395:1587.

Validity of the findings

The main findings shown in Figure 4 and Figure 6 illustrate the predicted R0 under different scenarios. However, as aforementioned, the contact matrices for each scenario were defined based on the percentage of particular job categories (derived from Office for National Statistics), but not the true contact rates in each occupation, which is unavailable from the BBC Pandemic database. Therefore, I am skeptical about the prediction.

Also, the present algorithm does not take confounding factors into account. These factors include, but not limit to, universal masking and social distancing policies, which have been extensively implemented during the pandemic. Such interventions can actually protect workers even if they have contacts while working, and thus affect R0 in reality.

Additional comments

In sum, I am concerned with applying BBC Pandemic contact matrices to the scenarios comprised of returning particular occupation workers to normal, as well as other policies that can confound the prediction. Because this is a data-driven study, its prediction would be biased if the training database were inadequate to answer the research question.

Reviewer 2 ·

Basic reporting

.

Experimental design

.

Validity of the findings

.

Additional comments

My major concern is that although the authors propose many detailed methods for the contact matrix and its link to Ro calculator, which as they acknowledged is already existing knowledge, in this results, they simply presented the results directly from Cori et al's approach (by running the 'EpiEstim' R package), which appears less novel.
My minor concern is that how do the authors consider the Rt estimated by using Cori et al's approach as Ro?
As such, I find fewer interests in this manuscript in either theoretical or epidemiological aspects. Thus, unless the authors could justify the novelty of this work, I do not recommend it for publishing.

---

## Round 0.2 · Major Revisions

Please, see the comments highlighted by the reviewer #1. These are key points for the scientific validity of your work.

Reviewer 1 ·

Basic reporting

The authors have improved basic reporting to my satisfaction.

Experimental design

The authors have added more details on the assumptions for algorithm building, which makes the methodology more clear. However, I am still uncomfortable with the strong assumptions. Most importantly, because this is a data-driven study, if there are essential data unavailable (such as an accurate occupation categorization to define essential workers and the distinct contact matrix for specific occupation instead of the homogeneous contact assumption), the prediction will be limited to be applied to the real world. I acknowledge the authors' novelty on utilizing contact matrix to predict COVID-19 transmission under different scenarios, but it seems to me that the currently available datasets could hardly make precise predictions.

Validity of the findings

Because the quality of available datasets is not satisfactory for the purpose of this study, and the current study is a data-driven study, I am not convinced the findings are valid.

Additional comments

I would suggest the authors to look for more literature before making any strong assumptions. For example, instead of assuming every occupation has the same contact matrix, the authors could define specific weights to individual occupation's contact matrix based on existing literature to better reflect the real world.

Reviewer 2 ·

Basic reporting

I am satisfied with the current version, thanks.

Experimental design

I am satisfied with the current version, thanks.

Validity of the findings

I am satisfied with the current version, thanks.

Additional comments

I am satisfied with the current version, thanks.

---

## Round 0.3 · Minor Revisions

Still pending some changes indicated by one of the reviewers.

Reviewer 1 ·

Basic reporting

The revision is to my satisfaction now, but a few minor corrections should be made before it can be accepted for publication.

1. Line 208-210, "Althogh the results of SC1 suggest that Northern Ireland’s R0 value remains below 1. SC2 suggests R0 values of lower than 1 in all the trusts, however, WHSCT, with R0 = 0.94 shows a higher sensitivity to changes in contact numbers compared to other trusts."
Please correct the spelling 'Althogh' to 'Although', and consider rephrase this sentence for readability.

2. Line 199, "...considering homogeneous social contacts" and Line 228, "...assuming homogeneous social behaviour"
I thought the weights for the average contact numbers were also applied to North Ireland. If that is the case, the assumption should be corrected.

Experimental design

no comment

Validity of the findings

no comment

---

## Round 0.4 · accepted · Accept

All the reviewers' concerns have been correctly addressed.

Reviewer 1 ·

Basic reporting

no comment

Experimental design

no comment

Validity of the findings

no comment